# Solving Bi-Objective Vehicle Routing Problems with Driving Risk Consideration for Hazardous Materials Transportation

**Huo Chai** 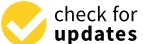**, Ruichun He \*, Ronggui Kang, Xiaoyan Jia and Cunjie Dai**

School of Traffic and Transportation, Lanzhou Jiaotong University, Lanzhou 730070, China; chaihuo@mail.lzjtu.cn (H.C.)
\* Correspondence: herc@mail.lzjtu.cn

**Abstract:** Driving behavior is an important factor affecting the risk of hazardous materials transportation. In this paper, we propose a transport risk evaluation method that considers driving risk. We consider driving risk and establish a model of vehicle routing problems with a soft time window for the transportation of hazardous materials and design a non-dominated genetic algorithm to solve the bi-objective optimization model. Taking a network of 23 nodes and 38 road segments as an example, 59 pareto-optimal solutions were obtained for six drivers on nine different paths. Comparing different solutions, it was found that driving risk, road population density, and transportation distance have different impacts on transport cost and risk. Choosing drivers and routes can adjust the propensity of cost and risk, allowing the decision-maker to select a solution for allocating drivers and routing vehicles according to their risk preference.

**Keywords:** driving risk; vehicle routing problem; bi-objective optimization; hazardous materials transportation

## 1. Introduction

With the ever-growing demand for transportation and the potential risk to public safety, transportation of hazardous materials has gained the attention of researchers and decision makers from government and non-governmental security organizations. It is estimated that four billion tons of hazardous materials are transported worldwide each year. According to the European Commission, around 60% of hazardous materials are transported by road in Europe [1]. In China, more than 92% of accidents related to hazardous materials occur during road transportation, and there are an average of 36 major accidents involving road transport of hazardous materials every year [2]. Despite the low accident rate, its impact on human and the environment is severe. Path optimization is an effective way to reduce the incidence and impact of accidents. Thus, it is of great research significance to optimize the path of hazardous materials transportation and provide a scientific and reasonable transport solution that decision makers can use to choose a route with an acceptable balance of cost and risk.

Vehicle routing problems for hazardous materials (HMVRP) is a type of bi-objective VRP that has been a focus by researchers for some time [3–5]. Tarantilis and Kiranoudis [6] proposed a list-based variant of the threshold accepting algorithm (LBTA) in order to minimize population exposure by solving a Capacitated VRP. Zografos and Androutsopoulos [7] defined the determination of hazardous materials distribution routes as a bi-objective vehicle routing problem with time windows; this is because risk minimization accompanies cost minimization in the objective function. They proposed a heuristic algorithm based on insertion to solve the problem and integrated it into the decision support system of hazardous materials transportation based on GIS. Androutsopoulos and Zografos [8] established a time-dependent bi-objective HMVRP with time windows and proposed a labeling algorithm based on the k-shortest path. Pradhananga et al. [9] designed a Multi-Objective Ant Colony

System (MOACS) to solve a bi-objective hazardous material routing problem in which they minimized the total scheduled travel time and the total risk value of the routes. Zheng [10] proposed a multi-objective optimization model for VRP in hazardous material transportation that aims to minimize road risk, the population affected along the way, and total cost while using a multi-objective genetic algorithm. Garrido et al. [11] considered the risk difference and risk equity of different types of hazardous materials, combined with social risk tolerance to optimize the routing, and formulated a multi-objective optimization that considers cumulative risk, cost, and travel time. Chai et al. [12] established mathematical models for vehicle routing and scheduling for logistical distribution of hazardous materials in full container loads (FCL) and converted the vehicle scheduling problem to a VRPTW. Cuneo et al. [13] formulated a capacitated VRPTW based on a case study related to the distribution of fuels for an oil company to its service stations. Bula et al. [14,15] proposed a multi-objective neighborhood search algorithm and a cost ε-constrained meta-heuristic algorithm for VRP of hazardous materials transportation. Meng et al. [16] proposed a multi-objective robust VRPTW model that considered the uncertainty of hazardous materials transportation and designed a hybrid evolutionary algorithm (HEA) to solve the robustness problem. Kang et al. [17] focused on the bi-objective shortest optimization problem (BSP) between single original-destination pairing of hazardous materials transportation that considered driving risk, which been evaluated by the analytic hierarchy process (AHP). Holeczek [18] assumed that the risk value decreases linearly with the load capacity, and considered the loading of hazardous materials in the risk assessment. Based on this, a VRPTW model was established and solved using a two-step method.

Most studies only consider the relationship between accident rate, residents, and environment in their risk evaluation, but driving risk and its impact are rarely considered. Aggressive behavior and bad driving habits are one of the main causes of traffic accidents. Based on the analysis of driving history data, driving behavior can be roughly described. Thus, it is necessary to consider driving risk in the risk evaluation of hazardous materials transportation. Driving risk is associated with many exposure factors [19], including driving skills [20], age [21], gender [22], driver behavior [23], etc. In addition, cumulative distance [24], road type [25], weather [26,27], and continuous driving time [28,29] are also contributory factors.

To make up for this defect in the existing literature, we propose to increase the consideration of driving risk when analyzing the risk of hazardous materials transportation. Based on driving risk evaluation, a Vehicle Routing Problem with a Soft Time Window for Hazardous Materials (HMVRPTW) model that aims to minimize risk and cost is established, and a Non-Dominated Genetic Algorithm II (NSGA-II) is designed to solve it. This model includes a coding rule. In addition, crossover and mutation strategies are proposed that can efficiently obtain the Pareto-optimal route.

The paper is organized as follows. Section 2 introduces the driving risk impact index and the driving risk evaluation. Section 3 establishes the bi-objective optimization model of HMVRPTW with the goal of minimizingrisk and cost. Section 4 discusses the design of an NSGA-II. Section 5 uses a hazardous materials transportation network, composed of 23 nodes and 38 links, as an example in order to obtain the Pareto-optimal route and analyze the different paths and driver assignments.

## 2. Driving Risk Evaluation

### 2.1. Driving Risk Impact Index

During the transportation of hazardous materials, driving risk primarily refers to the risk of crash, overturn, leakage, explosion, and other accidents that could occur during driving as a result of the driver's characteristics and driving habits. Driving data can directly reflect driving behavior, but it is meaningless to analyze the index of a single trip. It is necessary to analyze a large amount of historical driving data to comprehensively evaluate driving risk. According to the relevant research on hazardous materials transportation and other freight transportation, many indicators affect driving risk. Based on

the relevant transportation data of a regional branch of PetroChina in 2020, we chose the data of 100 drivers with more than three years of driving experience and selected the key indicators affecting driving risk from aspects including age, driving experience, educational background, gender, driving speed, driving habits, and heavy vehicle mileage. From this data, we built a driving risk evaluation index system, as shown in Table 1.

**Table 1.** Weight of all criteria for driving risk.

| Criterion | Weight | Plan (Level 1) | | Plan (Level 2) | |
|---|---|---|---|---|---|
| | | Criterion | Weight | Criterion | Weight |
| Driver's characteristics | 0.4425 | Age | 0.4039 | 25–30 | 0.1060 |
| | | | | 30–35 | 0.0665 |
| | | | | 35–40 | 0.1646 |
| | | | | 40–45 | 0.2698 |
| | | | | ≥45 | 0.3931 |
| | | Driving experience/years | 0.3404 | 3–7 | 0.4265 |
| | | | | 7–11 | 0.2537 |
| | | | | 11–15 | 0.1507 |
| | | | | 15–19 | 0.0867 |
| | | | | ≥19 | 0.0824 |
| | | Educational background | 0.1391 | Uneducated | 0.4743 |
| | | | | Primary | 0.2781 |
| | | | | Junior school | 0.1184 |
| | | | | High school | 0.0832 |
| | | | | Higher education | 0.0460 |
| | | Gender | 0.1166 | Male | 0.3333 |
| | | | | Female | 0.6667 |
| Driving habits | 0.5575 | Driving Speed/km·h$^{-1}$ (on urban roads) | 0.4182 | <45 | 0.0320 |
| | | | | 45–55 | 0.0583 |
| | | | | 55–65 | 0.1031 |
| | | | | 65–75 | 0.2976 |
| | | | | ≥75 | 0.5089 |
| | | Rapid acceleration/km·h$^{-2}$ | 0.1906 | <5 | 0.0330 |
| | | | | 5–10 | 0.0627 |
| | | | | 10–15 | 0.1401 |
| | | | | 15–20 | 0.2626 |
| | | | | ≥20 | 0.5016 |
| | | Rapid deceleration/km·h$^{-2}$ | 0.2707 | <5 | 0.0458 |
| | | | | 5–10 | 0.0907 |
| | | | | 10–15 | 0.1343 |
| | | | | 15–20 | 0.2515 |
| | | | | ≥20 | 0.5326 |
| | | Heavy vehicle mileage/km·d$^{-1}$ | 0.1205 | 120–150 | 0.1248 |
| | | | | 150–180 | 0.0778 |
| | | | | 180–210 | 0.3506 |
| | | | | >210 | 0.4918 |

### 2.2. Weights of Criterion

Different criteria have varying levels of influence on driving risk. By analyzing the driving data, our own investigation, the scoring of experts, and using the analytic hierarchy process (AHP) as an example, the assessment matrix of the criterion layer is determined by pairwise comparison of a driver's characteristics and driving habits. The satisfactory consistency of the results is calculated and checked to determine the weight of each criterion. The weights of all criteria at all levels are obtained by comparing them in pairs, as shown in Table 1.

Using the evaluation system, the driving risk weight of each driver is calculated to determine the transport risk associated with that driver during transportation of hazardous materials.

## 3. Problem Formulation

The HMVRPTW is defined on an undirected network $G = (N, A)$ comprised of the distribution center $o$, customer node subset $N_0$, and non-customer nodes set $N_1$. Each node $i \in N_0$ contains a fixed non-negative demand $q_i$, a known service time $\tau_i$, and a soft time window $[b_i, e_i]$. Each arc $(i, j) \in A$ is associated with a length $d_{ij}$, as well as any driver $m \in M$ driving a vehicle $k \in K$ transporting hazardous materials on a road segment. These road segments have an associated specified speed $v_{ijk}$, transport risk $r_{ijk}^m$, and cost $c_{ijk}^m$. The transport vehicles depart from the distribution center according to the capacity restrictions of the vehicles; after delivering to a certain number of customer nodes, they return to the distribution center. Each customer node can only be served by one vehicle and the vehicle can only pass once; for non-customer nodes, the vehicles can pass multiple times. The vehicles need to arrive at customer nodes within the specified time window; if they arrive early or late, the cost of transportation will be increased by a penalty. The transportation cost of different levels of drivers is also different. The core of the problem rests in how to arrange drivers and along which route to deliver so that the level of risk and target is optimal.

### 3.1. Assumption

(1) There is a linear relationship between transport risk and driving risk in hazardous materials transportation. (2) The loading of hazardous materials has a proportional effect on the consequences of vehicle accidents. (3) For any driver, the risk value of transportation with no load is zero. (4) Service time remains constant regardless of the vehicle load.

### 3.2. Symbols Definition

$G$: road network, $G = (N, A)$.
$N$: node set, $N = \{1, 2, 3, \ldots, n\}$.
$A$: link set among nodes, $A = \{(i, j) : i, j \in N\}$.
$o$: distribution center, $o \in N$.
$N_0$: customer nodes set, $N_0 \subset N$.
$N_1$: non-customer nodes set, $N_1 \subset N$.
$d_{ij}$: distance of link $(i, j) \in A$.
$K$: vehicle set.
$M$: driver set.
$q_i$: demand of customer $i$ $(i \in N_0)$.
$t_{ik}$: arrive time of vehicle $k$ $(k \in K)$ at node $i$ $(i \in N)$.
$\tau_i$: serve time at node $i$ $(i \in N_0)$.
$[b_i, e_i]$: time window restriction when node $i$ is serviced.
$l_{ok}$: the load of vehicle $k$ $(k \in K)$ from distribution center $o$ $(o \in N)$.
$l_{ik}$: the load of vehicle $k$ $(k \in K)$ arrives at node $i$ $(i \in N_0 \cup N_1)$.
$v_{ijk}$: driving speed of vehicle $k$ between $i$ and $j$ $((i, j) \in A)$
$w_m$: driving risk (weight) of driver $m$ $(m \in M)$.
$c_{ijk}^m$: cost of driver $m$ $(m \in M)$ drives vehicle $k$ $(k \in K)$ through link $(i, j) \in A$.
$r_{ijk}^m$: risk of driver $m$ $(m \in M)$ drives vehicle $k$ $(k \in K)$ through link $(i, j) \in A$.
$x_{ijk}^m$: decision variable. If driver $m$ $(m \in M)$ drives vehicle $k$ $(k \in K)$ through link $(i, j) \in A$, $x_{ijk}^m = 1$; else $x_{ijk}^m = 0$.

### 3.3. Risk and Cost

Base on the traditional risk evaluation [3], and taking into account the impact of loading capacity and driving risk according to assumptions (1) and (2), the risk of driver $m$ driving vehicle $k$ through link $(i, j)$ is:

$$r_{ijk}^{m} = d_{ij} p_{ij} \rho_{ij} \pi R^2 (w_m / w) \left( l_{jk} / l_{ok} \right) \tag{1}$$

where $p_{ij}$ is the accident probability of any vehicle passing through the link $(i, j)$, $\rho_{ij}$ is the population density around the link $(i, j)$, $R$ is the radius of the area affected by accidents, $w_m$ is the driving risk of driver $m$, and $w$ is the average value of all drivers' driving risks, which is calculated by:

$$w = \sum_{m \in M} w_m / |M| \tag{2}$$

The cost of transporting hazardous materials is mainly related to travel distance, fuel, road toll, material cost, maintenance cost, and labor cost. Taking these factors into account, the cost of driver $m$ driving vehicle $k$ with a load through link $(i, j)$ is:

$$c_{ijk}^{m} = (fs + ts + ms + cs + s_m) d_{ij} l_{jk} \tag{3}$$

and the cost of driver $m$ driving empty vehicle $k$ through link $(i, j)$ is:

$$c_{ijk}^{m} = (fs' + ts' + ms' + cs' + s_m) d_{ij} \tag{4}$$

For a vehicle carrying a load, $fs$ is the fuel consumption coefficient, $ts$ is the traffic cost coefficient, $ms$ is the material consumption coefficient, $cs$ is the maintenance cost coefficient, and $s_m$ is the labor cost coefficient, which is dependent on the driver $m$; for an empty vehicle, $fs'$ is the fuel consumption coefficient, $ts'$ is the traffic cost coefficient, $ms'$ is material consumption coefficient, and $cs'$ is the maintenance cost coefficient.

Labor cost is the salary of the driver and escorts. According to the relevant regulations of the logistics industry association, there must be one or more escorts accompanying the driver during hazardous materials transportation, excluding the driver themselves. Assuming that there is a linear relationship between the driver's salary and the driving risk, and that there is one escort accompanying the driver with a salary equal to the driver's, the labor cost $s_m$ of driver $m$ can be expressed as:

$$s_m = 2s_b (1 + (w_{\max} - w_m) / (w_{\max} - w_{\min})) \tag{5}$$

where $s_b$ is base salary of the freight logistics industry, and $w_{\max}$ and $w_{\min}$ are the upper and lower limits of driving risk for different drivers, respectively.

### 3.4. Mathematical Model

Considering the above costs and risks, the Vehicle Routing Problem with Soft Time Windows for Hazardous Materials Transportation (HMVRPTW) model is formulated as follows:

$$\min Z_1 = \sum_{m \in M} \sum_{k \in K} \sum_{(i,j) \in A} r_{ijk}^{m} x_{ijk}^{m} \tag{6}$$

$$\min Z_2 = \sum_{m \in M} \sum_{k \in K} \sum_{(i,j) \in A} c_{ijk}^{m} x_{ijk}^{m} + f_1 \sum_{k \in K} \sum_{i \in N_0} \max\{b_i - t_{ik}, 0\} + f_2 \sum_{k \in K} \sum_{i \in N_0} \max\{t_{ik} - e_i, 0\} \tag{7}$$

s.t.

$$\sum_{m \in M} \sum_{k \in K} \sum_{i \in N} x_{ijk}^{m} = 1 \quad \forall j \in N_0 \tag{8}$$

$$\sum_{m \in M} \sum_{k \in K} \sum_{j \in N} x_{ijk}^m = 1 \quad \forall i \in N_0 \tag{9}$$

$$\sum_{j \in N} x_{0jk}^m = 1 \quad \forall m \in M, k \in K \tag{10}$$

$$\sum_{i \in N} x_{i0k}^m = 1 \quad \forall m \in M, k \in K \tag{11}$$

$$\sum_{i \in N} x_{ijk}^m - \sum_{j \in N} x_{ijk}^m = 0 \quad \forall m \in M, k \in K \tag{12}$$

$$\sum_{m \in M} \sum_{j \in N_0} q_j \sum_{i \in N} x_{ijk}^m \le l_{ok} \quad \forall k \in K \tag{13}$$

$$\sum_{m \in M} \sum_{k \in K} \sum_{i \in N} q_j x_{ijk}^m = q_j \quad \forall j \in N_0 \tag{14}$$

$$x_{ijk}^m \in [0,1] \quad \forall m \in M, k \in K, (i,j) \in A \tag{15}$$

The model is a nonlinear integer programming problem in which objective function (6) minimizes the total risk and objective function (7) minimizes the cost. The model includes a soft time window constraint, with part 1 representing the transport cost in function (7) and part 2 and 3 representing the penalty cost for violating the soft time window constraint. $f_1$ and $f_2$ are the penalty coefficients for vehicles waiting and being late, respectively. Equations (8) and (9) guarantee that each customer node can only be served by one vehicle. Equation (8) states that the vehicle can only arrive at the customer node once; Equation (9) states that the vehicle can only depart from the customer node once. Equations (10)–(12) require that any driver driving the vehicle from the distribution center must return to the distribution center after completing the delivery. Equation (10) constrains the vehicle to start from node 0, while Equation (11) constrains the vehicle to return to node 0. Equation (12) ensures the conservation of inflow and outflow for any node. Equation (13) is the load constraint, ensuring that the demand at the customer node served by the vehicle shall not exceed the load at the departure of the vehicle from the distribution center. Equation (14) ensures that each customer node can only be served by one vehicle.

In function (7), the recursive relationship between the time $t_{jk}$ vehicle k arrives at a node $j$ and the time $t_{ik}$ it arrives at the previous node $i$ can be expressed as:

$$t_{jk} = \left( t_{ik} + \tau_i + d_{ij} / v_{ijk} \right) x_{ijk}^m \quad k \in K, (i,j) \in A \tag{16}$$

**4. Solution Methodology**

*4.1. NSGA-II Algorithm*

Genetic Algorithm (GA) is a computer science technique that uses the concept of simulated evolution to search for optimal solutions. It includes genetic mutation, adaptive selection, and crossover steps, through which the GA can find the optimal solution. In multi-objective optimization problem solving, Deb's Fast Non-Dominated Sorting Genetic Algorithm with Elite Strategy (NSGA-II) [30] is the most widely used. It employs fast non-dominated sorting to retain the superior individuals of the parent generation and introduce them directly into the offspring to prevent the loss of solution in the Pareto Front. It also proposes the crowding distance operator and elite strategy selection operator; this significantly reduces the computational complexity when compared to NSGA. The algorithm steps are as follows:

Step 1: Initialization. Randomly generate an initial population $P_0$ with population size $N$.

Step 2: Offspring generation. Use $P_i$ to select, cross, and mutate in order to obtain the offspring population $Q_i$ with population size $N$; set population $P_i$ and $Q_i$ and merge into $R_i$. Initially, $i = 0$.

Step 3: Fitness calculation. Decode the path according to the encoding rules and calculate the objective function.

Step 4: Update population. Use the elite strategy to generate the offspring; that is to say, the individuals in $R_i$ are non-dominated sorted and all their boundary sets $F = (F_1, F_2, \cdots)$ are constructed. The individuals are added to the offspring population $P_{i+1}$ from $F$ until it is size reaches $N$.

Step 5: Iteration. Go to Step 2 and repeat until the number of iterations $G$ is met.

### 4.2. Encoding and Decoding

Encoding and decoding are key technologies in the evolutionary algorithms; they directly affect the correctness and efficiency of the algorithm. To illustrate the individual encoding and decoding processes, take the transportation network shown in Figure 1 as an example. There are 10 nodes in the network shown in Figure 1; node 1 is the distribution center, the customer nodes are 5, 7, and 10, and the demand is 9 tons, 8 tons, and 13 tons respectively. The rated load of the vehicles is 20 tons.

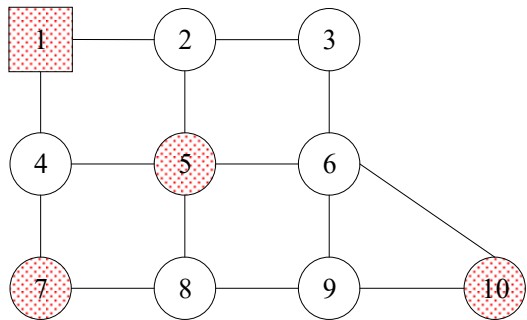

**Figure 1.** Example of the road network.

#### 4.2.1. Encoding

The integer coding method is used to encode the individual; the individual length is the number of nodes. The code contains two key pieces of information: the service order and the route of vehicles. The details of these pieces of information are explained in the decoding process.

#### 4.2.2. Decoding

The following are the decoding steps that are illustrated by taking the randomly generated individual 1-2-8-4-9-5-3-10-7-6 as an example.

**Step 1**: Service order

**Step 1.1**: Find customer service nodes in individuals.

**Step 1.2**: According to the demand of customer nodes and vehicle load constraints, the greedy strategy is adopted to select the customer nodes with its service order in the individual in order to obtain the customer nodes and service sequence.

In the example, the customer nodes are 5, 10, and 7. The customer nodes and order of vehicle service are shown in Table 2.

**Table 2.** Vehicle allocation.

| Vehicle | Customer Node | Service Order |
|---------|---------------|---------------|
| 1 | 5, 7 | 1→5→7→1 |
| 2 | 10 | 1→10→1 |

**Step 2**: Vehicle routing.

**Step 2.1**: Assign a new vehicle, determine the departing node (distribution center), and mark it in the individual.

**Step 2.2**: Traverse the individual to find the first node connected to the previous node, excluding the marked node.

**Step 2.3**: Judge whether there is a customer node in the connected nodes according to the vehicle service order. If so, select the customer node as the next node of the route and mark it, and remove the marking traces of other nodes in the individual except the customer node; if not, continue.

**Step 2.4**: According to the vehicle service order, judge whether there is an adjacent node shared by the current node and the customer node among the connected nodes. If so, select the point as the next node of the route and mark it according to the traversal order; if not, select and mark the next node according to the traversal order, and go to Step 2.2 until the vehicle has passed all customer nodes assigned to it and is ready to return to the distribution center.

**Step 2.5**: Find the node connected to the previous node (the last customer node of the vehicle distribution) in the individual.

**Step 2.6**: Judge whether there is a distribution center in the connected nodes (excluding the marked nodes). If so, the vehicle returns to the distribution center and the decoding is completed. If not, continue.

**Step 2.7**: Judge whether there is an adjacent node shared by the current node and distribution center 1 in the connected nodes. If so, select the node as the next node and mark it according to the traversal order; if not, select the next node and mark it according to the traversal order; go to step 2.6 until the vehicle returns to the distribution center.

**Step 2.8**: Repeat Step 2 to assign the driving path of the next vehicle until all vehicle driving paths are assigned.

Using the example individual as an example, allocate the driving path of the first vehicle. In the individual, non-customer nodes 2 and 4 are connected to node 1; they are the linked nodes of node 1 and customer node 5. Select node 2 as the next node and mark it. Nodes 3 and 5 are connected to node 2 according to the vehicle service order; select node 5 as the next node and mark it, then remove all of the markings (except for the customer node). The nodes connected to node 5 are nodes 2, 4, 6, and 8, which are all non-customer nodes. Among them, nodes 4 and 8 are the linked nodes of customer nodes 5 and 7, according to the traversal order; node 8 is selected as the next node and marked. The nodes connected with node 8 are nodes 7 and 9; node 7 is the customer node, so select it as the next node and mark it, then remove the mark trace (except for the customer node). The vehicle has passed all customer nodes assigned to it (nodes 5 and 7) and is ready to return to the distribution center. Nodes 4 and 8 are connected to node 7. Node 4 is the linked node of customer node 7 and distribution center 1; select it as the next node then mark it. The node connected to node 4 is distribution center 1; selecting it as the next node, the vehicle returns to the distribution center, so the distribution path of vehicle 1 is $1 \rightarrow 2 \rightarrow 5 \rightarrow 8 \rightarrow 7 \rightarrow 4 \rightarrow 1$. The above steps are shown in Figure 2. Repeat the decoding step to get the distribution path $1 \rightarrow 2 \rightarrow 3 \rightarrow 6 \rightarrow 10 \rightarrow 9 \rightarrow 6 \rightarrow 3 \rightarrow 2 \rightarrow 1$ of vehicle 2.

*4.3. Crossover Operator*

Two individuals are randomly selected from the population and the crossover operator is applied with a crossover probability to generate two new individuals. Take the two individuals in Figure 3 as an example to illustrate the crossover operation. The main steps are as follows:

**Step 1**: Randomly select the starting position $b$ and ending position $f$ of elements in individuals $p_1$ and $p_2$.

**Step 2**: Exchange selected elements in the two individuals to generate two new individuals $p_1'$ and $p_2'$.

**Step 3**: Establish the mapping between the selected elements in descending order. Taking two individual segments as an example. The mapping of the elements is shown in Figure 4.

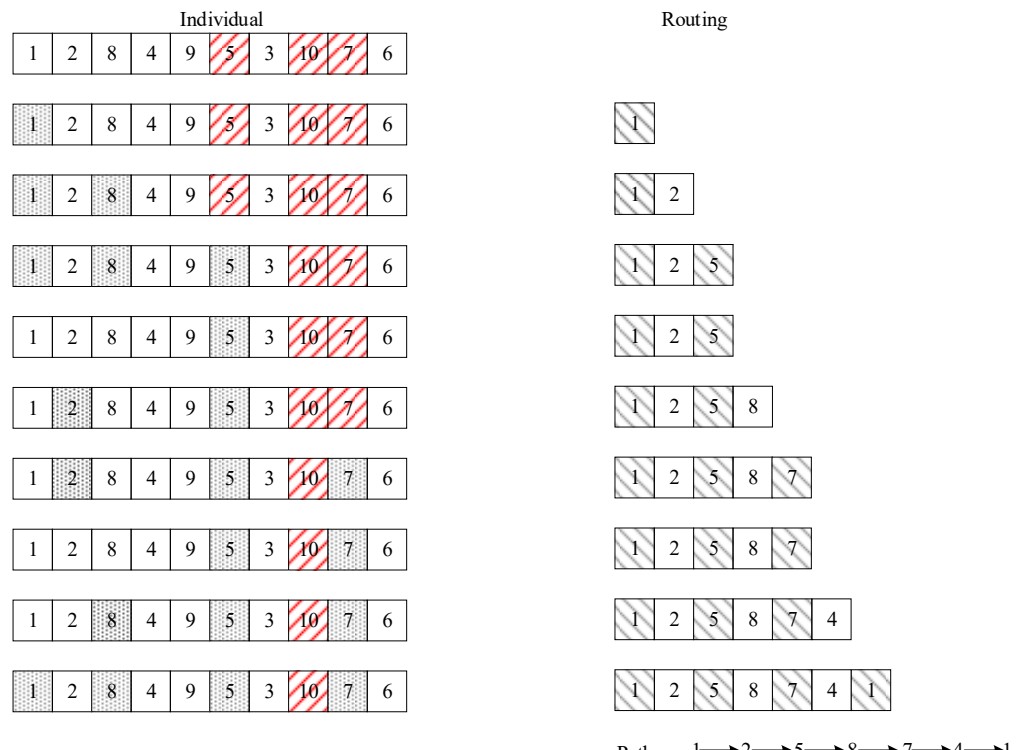

**Figure 2.** Decoding of vehicle 1 routing.

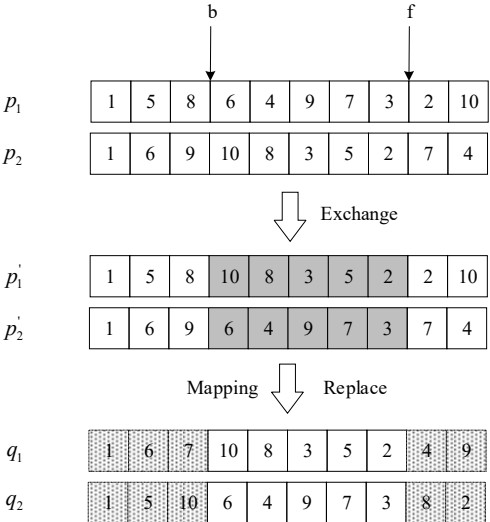

**Figure 3.** Example of crossover.

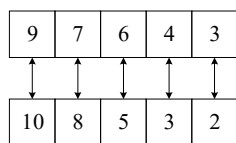

**Figure 4.** Mapping of the elements.

**Step 4**: Conflict detection to ensure that the elements in the new individual are free of conflict. For example, there are two element 2 s in the individual $p_1'$; there is conflict as a result. According to the mapping shown in Figure 4, the mapping $2 \leftrightarrow 3 \leftrightarrow 4$ between elements 4 and 2 is established (Figure 5), and the new individual $q_1$ has no conflict.

| 9 | 7 | 6 | 4 |
|---|---|---|---|
| 10 | 8 | 5 | 2 |

**Figure 5.** Final mapping of the elements.

**Step 5**: Replace the elements according to the final mapping relationship, then generate two new individuals $q_1$ and $q_2$.

### 4.4. Mutation Operator

Figure 6 shows the generation of a new individual through mutation. A randomly selected individual $p_3$ from the population with a mutation probability and two additional mutation positions are selected randomly. The elements in these two positions are then exchanged in pairs to generate a new individual $q_3$.

| $p_3$ | 1 | 3 | 6 | 7 | 8 | 4 | 9 | 10 | 2 | 5 |
|---|---|---|---|---|---|---|---|---|---|---|
| $q_3$ | 1 | 3 | 6 | 7 | 10 | 4 | 9 | 8 | 2 | 5 |

**Figure 6.** Example of mutation.

## 5. Numerical Experiments

### 5.1. Road Network

Take the road network composed of 23 nodes and 38 road segments (Figure 7) as an example, in which node 1 is the distribution center and nodes 10, 11, 14, 17, and 23 are the customer nodes; other nodes are non-customer nodes. Vehicles can travel on any road segment; the cost and risk are the same in both directions. The vehicle departs from the distribution center and returns after serving all customer nodes in turn. The rated load of the vehicle is 25 t, the radius of the area affected by a potential accident is 0.6 km, the driving speed of the vehicle is 40 km·h$^{-1}$, the waiting cost coefficient of vehicles is 60 CNY·h$^{-1}$, and the late cost coefficient is 90 CNY·h$^{-1}$. Table 3 shows the distance, the surrounding population density, and the accident probability of the road segment. Table 4 shows the demand information of the customer node.

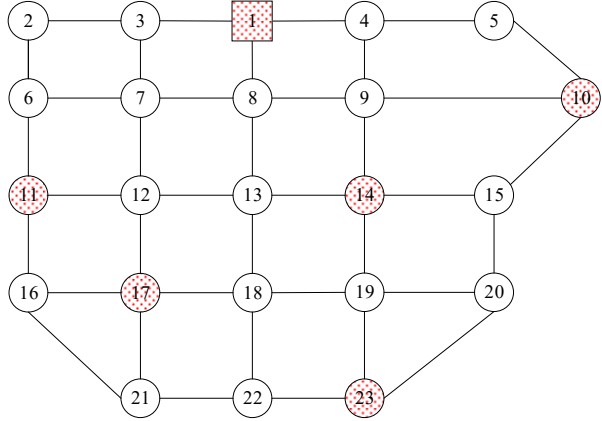

**Figure 7.** Road network.

**Table 3.** Road segment.

| ID | Road Segment | Distance/km | Population Density/Person·km$^{-2}$ | Accident Probability |
|---|---|---|---|---|
| 1 | 1-4 | 26.36 | 300 | 0.0042 |
| 2 | 1-8 | 25.80 | 400 | 0.0040 |
| 3 | 2-3 | 26.12 | 120 | 0.0031 |
| 4 | 2-6 | 28.40 | 75 | 0.0032 |
| 5 | 3-1 | 25.62 | 180 | 0.0034 |
| 6 | 3-7 | 21.29 | 360 | 0.0036 |
| 7 | 4-5 | 30.09 | 320 | 0.0062 |
| 8 | 4-9 | 25.09 | 290 | 0.0035 |
| 9 | 5-10 | 30.06 | 260 | 0.0065 |
| 10 | 6-11 | 30.89 | 200 | 0.0042 |
| 11 | 6-7 | 23.19 | 690 | 0.0040 |
| 12 | 7-12 | 24.14 | 660 | 0.0042 |
| 13 | 7-8 | 21.35 | 400 | 0.0044 |
| 14 | 8-9 | 25.12 | 480 | 0.0043 |
| 15 | 8-13 | 27.48 | 520 | 0.0047 |
| 16 | 9-10 | 40.20 | 400 | 0.0045 |
| 17 | 9-14 | 25.60 | 760 | 0.0049 |
| 18 | 10-15 | 32.50 | 430 | 0.0042 |
| 19 | 11-12 | 20.67 | 680 | 0.0054 |
| 20 | 11-16 | 35.60 | 300 | 0.0052 |
| 21 | 12-13 | 23.50 | 700 | 0.0045 |
| 22 | 12-17 | 30.20 | 620 | 0.0050 |
| 23 | 13-14 | 25.29 | 620 | 0.0050 |
| 24 | 13-18 | 35.55 | 710 | 0.0053 |
| 25 | 14-15 | 27.35 | 850 | 0.0049 |
| 26 | 14-19 | 36.78 | 720 | 0.0048 |
| 27 | 15-20 | 35.70 | 650 | 0.0055 |
| 28 | 16-17 | 21.10 | 550 | 0.0032 |
| 29 | 16-21 | 30.12 | 420 | 0.0060 |
| 30 | 17-18 | 28.50 | 640 | 0.0050 |
| 31 | 17-21 | 28.56 | 500 | 0.0040 |
| 32 | 18-19 | 28.30 | 630 | 0.0049 |
| 33 | 18-22 | 30.58 | 530 | 0.0052 |
| 34 | 19-20 | 27.08 | 640 | 0.0036 |
| 35 | 19-23 | 24.16 | 480 | 0.0042 |
| 36 | 20-23 | 29.46 | 160 | 0.0064 |
| 37 | 21-22 | 32.60 | 200 | 0.0058 |
| 38 | 22-23 | 30.60 | 240 | 0.0055 |

**Table 4.** Customer node.

| Node | Demand/t | Service Duration/h | Time Window |
|---|---|---|---|
| 10 | 10 | 1 | 09:30–11:00 |
| 11 | 8 | 1 | 13:00–14:30 |
| 14 | 10 | 1 | 10:30–12:00 |
| 17 | 7 | 1 | 14:30–16:30 |
| 23 | 8 | 1 | 12:30–14:00 |

*5.2. Driving Risk and Cost*

Taking the gasoline, materials, and salary in China as cost parameters, the fuel consumption coefficient when loaded is $fs = 0.2309$ CNY·t$^{-1}$·km$^{-1}$, the material consumption coefficient is $ms = 0.0212$ CNY·t$^{-1}$·km$^{-1}$, and maintenance cost coefficient is $cs = 0.0093$ CNY·t$^{-1}$·km$^{-1}$. When the vehicle is unloaded, the fuel consumption coefficient is $fs' = 1.5687$ CNY·km$^{-1}$, the material consumption coefficient is $ms' = 0.1673$ CNY·km$^{-1}$,

and the maintenance cost coefficient is $cs' = 0.0426$ CNY·km$^{-1}$. Based on the driver's historical data, the driver's driving risk is $w_m \in [0.1300, 0.2600]$, and the labor cost is $s_m \in [0.07, 0.12]$ (CNY·t$^{-1}$·km$^{-1}$). Table 5 shows the driving risk value and cost information of the six drivers in this example ($w = 0.1942$).

**Table 5.** Driving risk and cost of six drivers.

| Driver | 1 | 2 | 3 | 4 | 5 | 6 |
|---|---|---|---|---|---|---|
| Driving risk ($w_m$) | 0.2129 | 0.1849 | 0.2417 | 0.1370 | 0.1622 | 0.2264 |
| $w_m/w$ | 1.0965 | 0.9522 | 1.2447 | 0.7055 | 0.8352 | 1.1659 |
| Driving cost/CNY·t$^{-1}$·km$^{-1}$ | 0.0881 | 0.0989 | 0.0770 | 0.1173 | 0.1076 | 0.0829 |

### 5.3. Solution by NSGA-II

The algorithm was programmed in MATLAB (r2016a) with the following parameter settings: population size $N = 600$, maximum iteration times $G = 200$, crossover probability 0.8, and mutation probability 0.2. The parameter values in the algorithm are set according to the problem size and the analysis in the literature [30]. The Pareto-optimal front, which includes 59 driving solutions on nine different paths obtained by the program, is shown in Figure 8, and the details of the nine different paths are shown in Table 6.

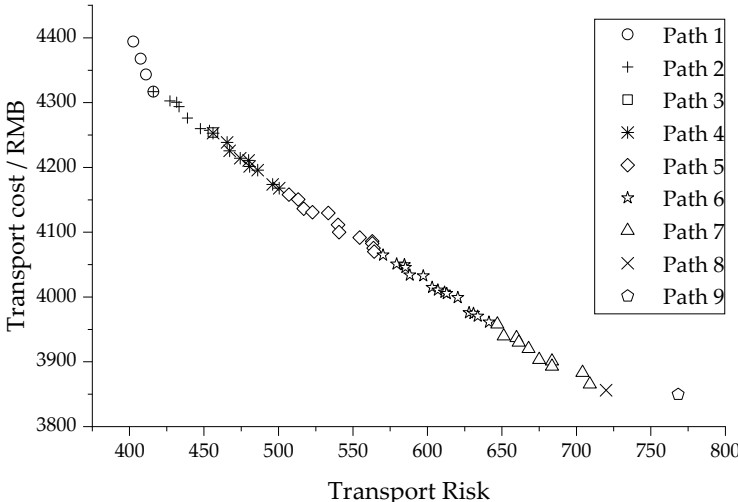

**Figure 8.** Pareto-optimal solution set.

The program running environment is i7-8550U/8G, calculating 10 times in total, with an average running time of 7.13 s; each time can obtain a Pareto optimal solution. In order to verify the superiority of the algorithm proposed in this paper to solve the problem, an ant colony algorithm is designed according to the literature [9]. When the running time of the ant colony algorithm (500 ants and 30 iterations) reaches 7.13 s, the population terminates the iteration and the solution obtained from one run is compared with the Pareto optimal solution obtained by the algorithm proposed in this paper; these comparisons show that the Pareto front obtained by the algorithm proposed in this paper is obviously better (see Figure 9). By running the ant colony algorithm ten times with the 80% overlap of the solution set obtained by the algorithm proposed in this paper as the ending condition, the average run time was 13.38 s. It is evident that the algorithm proposed in this paper has a significant advantage in solving the problem when compared to the ant colony algorithm.

**Table 6.** Details of the nine paths.

| Path No. | Vehicle | Path | Distance/km |
|---|---|---|---|
| 1 | 1 | 1→4→9→14→13→12→11→12→17→12→7→3→1 | 624.09 |
| | 2 | 1→4→5→10→15→20→23→19→18→13→8→1 | |
| 2 | 1 | 1→4→9→14→13→12→11→16→17→12→7→3→1 | 629.92 |
| | 2 | 1→4→5→10→15→20→23→19→18→13→8→1 | |
| 3 | 1 | 1→4→9→14→13→12→11→16→17→12→7→3→1 | 635.06 |
| | 2 | 1→4→9→10→15→20→23→19→18→13→8→1 | |
| 4 | 1 | 1→4→9→14→19→23→19→18→17→12→7→3→1 | 639.18 |
| | 2 | 1→4→5→10→9→8→7→6→11→12→7→3→1 | |
| 5 | 1 | 1→4→9→14→19→23→22→21→17→12→7→3→1 | 649.98 |
| | 2 | 1→4→5→10→9→8→7→6→11→12→7→3→1 | |
| 6 | 1 | 1→4→9→14→19→23→19→18→17→12→7→3→1 | 629.91 |
| | 2 | 1→4→5→10→9→8→7→12→11→12→7→3→1 | |
| 7 | 1 | 1→4→9→14→19→23→22→21→17→12→7→3→1 | 640.71 |
| | 2 | 1→4→5→10→9→8→7→12→11→12→7→3→1 | |
| 8 | 1 | 1→8→9→14→19→23→19→18→17→12→7→3→1 | 629.38 |
| | 2 | 1→4→5→10→9→8→7→12→11→12→7→3→1 | |
| 9 | 1 | 1→4→9→14→19→23→22→21→17→12→7→3→1 | 645.85 |
| | 2 | 1→4→9→10→9→8→7→6→11→12→7→3→1 | |

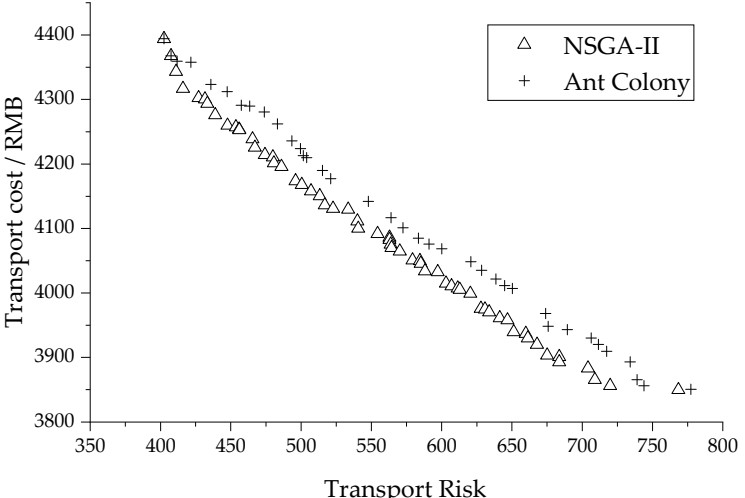

**Figure 9.** Pareto-optimal solution set obtained by NSGA-II and Ant Colony.

*5.4. Results and Discussion*

Table 7 shows the Pareto-optimal solutions with their associated risk and cost. Solution 1 has the lowest total risk; it is transported by drivers 4 and 5 through path 8. Solution 59 has the lowest total cost; it is transported by drivers 3 and 6 through path 3. Both routes are not the shortest distance.

Different drivers are selected on the same path, and the total risk and cost are different. Taking path 5 as an example, different drivers are selected. The Pareto-optimal solution is shown in Table 8.

**Table 7.** Pareto-optimal solutions (partial).

| Solution No. | Path | Driver | Risk | Cost /CNY |
|---|---|---|---|---|
| 1 | $1 \to 8 \to 9 \to \underline{14} \to 19 \to \underline{23} \to 19 \to 18 \to \underline{17} \to 12 \to 7 \to 3 \to \underline{1}$ | 4 | 402.5062 | 4394.09 |
| | $\underline{1} \to 4 \to 5 \to \underline{10} \to 9 \to 8 \to 7 \to 12 \to \underline{11} \to 12 \to 7 \to 3 \to \underline{1}$ | 5 | | |
| 59 | $\underline{1} \to 4 \to 9 \to \underline{14} \to 19 \to \underline{23} \to 22 \to 21 \to \underline{17} \to 12 \to 7 \to 3 \to \underline{1}$ | 3 | 768.3525 | 3849.75 |
| | $\underline{1} \to 4 \to 9 \to \underline{10} \to 9 \to 8 \to 7 \to 6 \to \underline{11} \to 12 \to 7 \to 3 \to 1$ | 6 | | |

**Table 8.** Pareto-optimal solutions (partial).

| Solution No. | Path | Driver | Risk | Cost /CNY |
|---|---|---|---|---|
| 3 | $\underline{1} \to 4 \to 9 \to \underline{14} \to 19 \to \underline{23} \to 22 \to 21 \to \underline{17} \to 12 \to 7 \to 3 \to \underline{1}$ | 4 | 411.1256 | 4343.33 |
| | $\underline{1} \to 4 \to 5 \to \underline{10} \to 9 \to 8 \to 7 \to 6 \to \underline{11} \to 12 \to 7 \to 3 \to \underline{1}$ | 5 | | |
| 13 | $\underline{1} \to 4 \to 9 \to \underline{14} \to 19 \to \underline{23} \to 22 \to 21 \to \underline{17} \to 12 \to 7 \to 3 \to \underline{1}$ | 4 | 465.5105 | 4238.69 |
| | $\underline{1} \to 4 \to 5 \to \underline{10} \to 9 \to 8 \to 7 \to 6 \to \underline{11} \to 12 \to 7 \to 3 \to \underline{1}$ | 1 | | |
| 26 | $\underline{1} \to 4 \to 9 \to \underline{14} \to 19 \to \underline{23} \to 22 \to 21 \to \underline{17} \to 12 \to 7 \to 3 \to \underline{1}$ | 5 | 539.9801 | 4111.56 |
| | $\underline{1} \to 4 \to 5 \to \underline{10} \to 9 \to 8 \to 7 \to 6 \to \underline{11} \to 12 \to 7 \to 3 \to \underline{1}$ | 3 | | |
| 30 | $\underline{1} \to 4 \to 9 \to \underline{14} \to 19 \to \underline{23} \to 22 \to 21 \to \underline{17} \to 12 \to 7 \to 3 \to \underline{1}$ | 1 | 562.9320 | 4082.62 |
| | $\underline{1} \to 4 \to 5 \to \underline{10} \to 9 \to 8 \to 7 \to 6 \to \underline{11} \to 12 \to 7 \to 3 \to \underline{1}$ | 6 | | |
| 41 | $\underline{1} \to 4 \to 9 \to \underline{14} \to 19 \to \underline{23} \to 22 \to 21 \to \underline{17} \to 12 \to 7 \to 3 \to \underline{1}$ | 6 | 611.4671 | 4007.39 |
| | $\underline{1} \to 4 \to 5 \to \underline{10} \to 9 \to 8 \to 7 \to 6 \to \underline{11} \to 12 \to 7 \to 3 \to \underline{1}$ | 3 | | |
| 49 | $\underline{1} \to 4 \to 9 \to \underline{14} \to 19 \to \underline{23} \to 22 \to 21 \to \underline{17} \to 12 \to 7 \to 3 \to \underline{1}$ | 3 | 651.2104 | 3939.51 |
| | $\underline{1} \to 4 \to 5 \to \underline{10} \to 9 \to 8 \to 7 \to 6 \to \underline{11} \to 12 \to 7 \to 3 \to \underline{1}$ | 6 | | |

Table 8 shows that the driving risks of six drivers are ranked from low to high, with driver 4 having the lowest driving risk and driver 3 having the highest driving risk. Table 8 shows that for transportation on the same path, the decision maker can arrange drivers according to the preference of risk and cost. (1) When focusing on the risk, the decision maker needs to choose drivers with low driving risk. For example, when compared with solution 49, the driving risk of drivers 4 and 5 are lower than that of drivers 3 and 6, and the total risk of solution 3 is lower than that of solution 49. (2) When focusing on cost, decision makers can choose drivers with higher driving risk. For example, compared with solution 13, the driving risks of drivers 6 and 3 in solution 41 are greater than those of drivers 4 and 1, and the total cost of solution 41 is more economical than that of solution 13. (3) When comprehensively considering the risk and cost, the decision-maker can combine the drivers with both low and high driving risk. For example, when compared with solutions 3 and 41, the total cost of solution 26 is lower than that of solution 3 and the total risk is lower than that of solution 49.

Under the same service order, the total risk and cost of the same drivers on different paths can also differ greatly. Taking drivers 4 and 5 as an example, Table 9 shows the total risk and cost of different paths under the same service order.

Table 9 shows that the same drivers are selected to transport in the same service order. If focused on the risk, the decision maker can reduce the risk by selecting the road segment with a relatively low surrounding population density. For example, the population density of road segments (11→16) and (16→17) in solution 2 are lower than those of road segments (11→12) and (12→17) in solution 7. If focused on the cost, the cost can be reduced by selecting the road segments with short distances. For example, if the distance of the path in solution 7 is shorter than that of solution 2, the cost will be lower.

**Table 9.** Pareto-optimal solutions (partial).

| Solution No. | Path | Driver | Risk | Cost /CNY |
|---|---|---|---|---|
| 2 | 1→4→9→14→13→12→11→16→17→12→7→3→1 | 4 | 407.4593 | 4367.60 |
|  | 1→4→9→10→15→20→23→19→18→13→8→1 | 5 |  |  |
| 4 | 1→4→9→14→13→12→11→16→17→12→7→3→1 | 4 | 416.0787 | 4316.85 |
|  | 1→4→5→10→15→20→23→19→18→13→8→1 | 5 |  |  |
| 7 | 1→4→9→14→13→12→11→12→17→12→7→3→1 | 4 | 433.2476 | 4293.67 |
|  | 1→4→5→10→15→20→23→19→18→13→8→1 | 5 |  |  |

For different drivers driving on different paths, take solutions 5, 6, and 7 as examples; if different drivers are selected, the total risk and cost are shown in Table 10.

**Table 10.** Pareto-optimal solutions (partial).

| Solution No. | Path | Driver | Risk | Cost /CNY |
|---|---|---|---|---|
| 3 | 1→4→9→14→19→23→22→21→17→12→7→3→1 | 4 | 411.1256 | 4343.33 |
|  | 1→4→5→10→9→8→7→6→11→12→7→3→1 | 6 |  |  |
| 13 | 1→4→9→14→19→23→19→18→17→12→7→3→1 | 4 | 465.5105 | 4238.69 |
|  | 1→4→5→10→9→8→7→12→11→12→7→3→1 | 1 |  |  |
| 36 | 1→4→9→14→19→23→22→21→17→12→7→3→1 | 2 | 585.2382 | 4045.60 |
|  | 1→4→5→10→9→8→7→12→11→12→7→3→1 | 6 |  |  |
| 38 | 1→4→9→14→19→23→19→18→17→12→7→3→1 | 2 | 597.1895 | 4032.59 |
|  | 1→4→5→10→9→8→7→12→11→12→7→3→1 | 1 |  |  |

Table 10 shows that there is a situation where drivers with high driving risk can be used to transport, but that the total risk is small; this is caused by different paths having different population densities. For example, between solutions 3 and 13, the driving risk of driver 6 is higher than that of driver 1, but the total risk of solution 3 is lower than that of solution 13 due to driver 1 driving the vehicle through high-density population areas (23→19), (19→18), and (18→17). The same phenomenon also occurs between solutions 36 and 38. Therefore, when the decision maker chooses drivers with high driving risk to transport hazardous materials with cost as the motivating focus, they can reduce the risk by choosing those road segments with relatively low population density.

In conclusion, if the total risk needs to be reduced, the decision-maker should choose drivers with less driving risk and paths with low population density. If the cost needs to be reduced, drivers with high driving risk can be selected, but the risk can be reduced by selecting paths with a low-density population. If the risk and cost need to be considered at the same time, the decision maker should choose the driver with less driving risk and those paths with shorter distances.

## 6. Conclusions and Future Work

This study has considered driving risk in the hazardous materials vehicle routing problem, doing the following work in terms of risk assessment, and presenting a mathematical model to solution algorithm:

(1) For the evaluation of driving risk in hazardous materials transportation, an AHP method is proposed that can evaluate driving risk according to the history of driving data, the investigation, and scoring by experts.

(2) For the mathematical model, the coexistence of customer nodes and non-customer nodes in the transportation network is considered in the HMVRPTW model. For the general VRP problems, non-customer nodes are eliminated by solving the shortest path between two customer nodes. However, in hazardous materials transportation, there is a set of Pareto-optimal shortest paths between any two customer nodes; these paths may pass through different non-customer nodes.

(3) For the solving method, an individual encoding/decoding method suitable for the mathematical model in this study is proposed. This method includes both the service order of the customer nodes and all the nodes that the vehicle passes through. For this coding method, an evolutionary strategy including crossover and mutation is proposed.

The proposed approaches, when applied to the distribution of hazardous materials, generate Pareto-optimal solutions from which the decision maker can select the most appropriate driver allocation and vehicle routing, based on their risk preference.

Similar to the transport risk of hazardous materials, there is no unique or recognized correct standard for the evaluation of driving risk. The AHP method suggested by experts has been used in the study; this method has a certain subjectivity, because the objectivity of the evaluation results of driving risk need to be further verified. Thus, in future research, the number of samples will be expanded to obtain a more applicable driving risk evaluation model to more adequately study the impact of driving risk on transport risk and routing optimization for hazardous materials transportation. The VRP problem model of hazardous materials transportation considering the driving risk is established and a solving method is proposed. However, to build software applications on this basis, there may be a long way to go in terms of data collection, algorithm optimization and result verification.

**Author Contributions:** Conceptualization, H.C. and R.H.; methodology, H.C. and R.K.; software, R.K.; validation, H.C. and R.K.; formal analysis, H.C.; investigation, C.D.; resources, C.D.; data curation, H.C.; writing—original draft preparation, R.K.; writing—review and editing, H.C. and X.J.; supervision, R.H.; project administration, H.C.; funding acquisition, R.H. All authors have read and agreed to the published version of the manuscript.

**Funding:** This research was funded by the National Natural Science Foundation of China, grant number 71961015 and grant number 52162041, "Double-First Class" Major Research Programs, Educational Department of Gansu Province grant number GSSYLXM-04.

**Institutional Review Board Statement:** Not applicable.

**Informed Consent Statement:** Not applicable.

**Data Availability Statement:** The data generated or analyzed during this study are included in this published article.

**Acknowledgments:** We would like to thank the anonymous referees for their insightful comments that helped us to improve the quality of this paper.

**Conflicts of Interest:** The authors declare no conflict of interest. The funders had no role in the design of the study; in the collection, analyses, or interpretation of data; in the writing of the manuscript; or in the decision to publish the results.

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
