# Peer review of "Solving Bi-Objective Vehicle Routing Problems with Driving Risk Consideration for Hazardous Materials Transportation"

_sustainability, doi:10.3390/su15097619_

Round 1

Reviewer 1 Report (Previous Reviewer 1)

There is a lot of changes in the manuscript file, so It is hard to check it now. 

It is strange that the authors do not want to put a link to their previous Chinese article, if they provided a link to it with English title and add "(in Chinese)", then someone from this country could find the article and read it. Do not afraid to provide links to the previous research even in small journals, they are your achievements and  the research direction should be clearly stated. 

I think that the authors had done a good work, changed a lot from the very first version here, and deserve the article to be published.

Almost fine. 

Too much passive tense in Conclusion. 

Author Response

Thanks for your valuable comments on our manuscript, which is very helpful for us to improve our study. The following are the formal reply to the questions raised , and the revised content were marked with green and through the “tracked changes” in Word. We polished the full text under the guidance of a native-English language teacher, and revised many syntaxes error and non-idiomatic expressions, including the Conclusion you pointed out.

Reviewer 2 Report (Previous Reviewer 3)

The authors made some changes (which were not requested by me). However, I maintain my opinion, namely acceptance for publication.

Author Response

Thank you for all you have done for our manuscript.

Reviewer 3 Report (New Reviewer)

1. Till the title is not clear.

2. The abstract can be rewritten by adding some numerical results.  3. The paper needs language editing.  Example:   In China, more than 92% of accidents related to hazardous materials related  occur during road transportation, and there are an average of 36 major accidents involving road transport of hazardous materials every year [2] 4. Please provide a detailed literature review. Holeczek [1718] considered the loading of hazardous materials in the risk assessment and established the VRPTW model.  - What is the inference from this? 5. Is there any reference for Table 1? If so, provide the reference. Otherwise, explain how did you assign the weights? 6. In section 3, first write the problem definition. Then the model, assumptions, etc. 7. More explanation is required to understand the constraints. 8. Change the title of section 4.  9. Provide a simple description of basic GA. 10. The performance of any metaheuristics mainly depends on the parameters. Please explain the parameters used in the current work.  12.  The proposed algorithm may be compared with other algorithms proposed in the literature.  13. The details of computational time are not available. Please provide this as it is an important one.    14. The proposed algorithm may be applied to a real case problem. 

The paper needs language editing. 

Author Response

Thanks for your valuable comments on our manuscript, which is very helpful for us to improve our study. The following are the formal reply to the questions raised , and the revised content were marked with green and through the “tracked changes” in Word.

1. Till the title is not clear.

Response 1: After careful consideration, we have modified the title of the paper to “Solving bi-objective vehicle routing problems with driving risk consideration for hazardous materials transportation”.

2. The abstract can be rewritten by adding some numerical results.

Response 2: Based on your suggestion, we have rewritten the conclusion section of the abstract.

3. The paper needs language editing. Example: In China, more than 92% of accidents related to hazardous materials related occur during road transportation, and there are an average of 36 major accidents involving road transport of hazardous materials every year [2].

Response 3: We polished the full text under the guidance of a native-English language teacher, and revised many syntaxes error and non-idiomatic expressions, including the sentence you pointed out.

4. Please provide a detailed literature review. Holeczek [1718] considered the loading of hazardous materials in the risk assessment and established the VRPTW model. - What is the inference from this?

Response 4: We are sorry for not being able to clarify the contribution of reference [18]. After revision, we have supplemented the innovative points of reference [18] in the HVRP.

5. Is there any reference for Table 1? If so, provide the reference. Otherwise, explain how did you assign the weights?

Response 5: Table 1 presents the indicators developed from the randomly selected 100 drivers at a regional branch of PetroChina. Expert scores were used and the Analytic Hierarchy Process (AHP) was employed to build a driving risk evaluation index system and determine the influence weights of different factors on driver driving risk.

6. In section 3, first write the problem definition. Then the model, assumptions, etc.

Response 6: According to your suggestion, we added the basic definition of the problem at the beginning of 3. Problem formulation, and then expanded the details of the model from assumptions, variable definitions, etc.

7. More explanation is required to understand the constraints.

Response 7: Following the model, we wrote comments on each equation in the model, including the objective function and constraints.

8.Change the title of section 4.

Response 8: The title of Section 4 was changed from "Problem Solving" to "Solution methodology", and other titles were also checked. The title of Section 3 was changed from "HVRPTW model" to "Problem formulation".

9. Provide a simple description of basic GA.

Response 9:  Before introducing the NSGA-II algorithm, a brief description of the basic genetic algorithm was added.

10. The performance of any metaheuristics mainly depends on the parameters. Please explain the parameters used in the current work.

Response 10:  The parameter values in the algorithm are set according to the problem size and the analysis in the literature [30]. We added this information in the revised text.

(No comment number 11 was found.)

12. The proposed algorithm may be compared with other algorithms proposed in the literature. 

Response 12: In this revision, we designed an ant colony algorithm according to the literature [9], and calculated the numerical example in the paper again with the ant colony algorithm and NSGA-II, and compared the Pareto optimal convergence and the program running time. The comparison result of one comparison is shown in Figure 9. At the same time, it was also found that the problem that some of the solutions are not displayed in Figure 8 due to the wrong setting of the coordinate axis range was solved.

13. The details of computational time are not available. Please provide this as it is an important one.

Response 13: This problem was addressed as part of the modification of Problem 12.

14. The proposed algorithm may be applied to a real case problem.

Response 14: This is a difficult problem for us now. One of the items in our project is to apply our research to a real problem. The research case we chose is Lanzhou where we are located, but at present only preliminary data preparation (network and distance) is available, and population distribution is still being collected, so this study can only design a case to verify our model and method.

Round 2

Reviewer 3 Report (New Reviewer)

Congratulations. The quality of the paper is improved now. 

This manuscript is a resubmission of an earlier submission. The following is a list of the peer review reports and author responses from that submission.

Round 1

Reviewer 1 Report

I would reject the paper again,since the authors gave the answers about their previous publication in the letter but did nothing to include such statements in the text of the manuscript.

The main research direction followed by the authors, all the related contributions should be given in the text (in the Introduction or a special section), and the differencies between previous findings and the results should be clearly stated ("in contrast, we have used such methods as...and we found out that but now we try to enrich the research because...").

So, please include text of your letter to me into the article, provide all the links to the previous papers and tell us why are you doing current research and state its place in the whole research direction. 

Also, Table 1 can be better presented and (8)-(15) are better wrap with text for each formula with explanations. 

5.4 Result and discussion => 5.4 Results and discussions

Author Response

I would reject the paper again, since the authors gave the answers about their previous publication in the letter but did nothing to include such statements in the text of the manuscript.

Point 1:

The main research direction followed by the authors, all the related contributions should be given in the text (in the Introduction or a special section), and the differencies between previous findings and the results should be clearly stated ("in contrast, we have used such methods as...and we found out that but now we try to enrich the research because...").

So, please include text of your letter to me into the article, provide all the links to the previous papers and tell us why are you doing current research and state its place in the whole research direction. 

Response 1:

Thank you for your continuous guidance on our study. Previously, we had been focusing on explaining the differences between our two papers/manuscripts, but neglected to cite them in new research. Instead, we added the citations as [17] in the manuscript. In fact, we did not cite it because we assumed that citing Chinese papers was often beyond the reach of readers.

Point 2:

Also, Table 1 can be better presented and (8)-(15) are better wrap with text for each formula with explanations. 

Response 2:

Table 1 has been adjusted to remove unnecessary columns and inner borders, and each formula that appears has been explained separately.

Point 3:

5.4 Result and discussion => 5.4 Results and discussions

Response 3:

It has been modified.

Reviewer 2 Report

I reviewed the new version of the paper, which was scheduled for a major revision. However, the changes are minimal: a few words have been added in the text, as well as some further specification in the conclusions. Furthermore, in replying to the previous comments, the authors limited themselves to addressing the similarity with another article, without considering the further aspects that I had indicated.

Reviewer 3 Report

The authors have made all the corrections requested by me, so I recommend acceptance.

Reviewer 4 Report

Driving behavior is a very interesting topic in the research area. The revised paper is well organized and written. Since the paper is about driving behavior, the paper could be further improved by considering more related work on driving states estimation. Please elaborate more on the literature review section from the configuration of the sensor: yolov5-tassel: detecting tassels in rgb uav imagery with improved yolov5 based on transfer learning, improved vehicle localization using on-board sensors and vehicle lateral velocity, autonomous vehicle kinematics and dynamics synthesis for sideslip angle estimation based on consensus kalman filter, automated driving systems data acquisition and processing platform.